# NeuroDNAAI: Neural Pipeline Approaches for the Advancing DNA-Based Information Storage as a Sustainable Digital Medium Using Deep Learning Framework

## Abstract

DNA is a promising medium for digital information storage for its exceptional density and durability. While prior studies advanced coding theory, workflow design, and simulation tools, challenges such as synthesis costs, sequencing errors, and biological constraints (GC-content imbalance, homopolymers) limit practical deployment. To address this, our framework draws from quantum parallelism concepts to enhance encoding diversity and resilience, integrating biologically informed constraints with deep learning to enhance error mitigation in DNA storage. NeuroDNAAI encodes binary data streams into symbolic DNA sequences, transmits them through a noisy channel with substitutions, insertions, and deletions, and reconstructs them with high fidelity. Our results show that traditional prompting or rule-based schemes fail to adapt effectively to realistic noise, whereas NeuroDNAAI achieves superior accuracy. Experiments on benchmark datasets demonstrate low bit error rates for both text and images. By unifying theory, workflow, and simulation into one pipeline, NeuroDNAAI enables scalable, biologically valid archival DNA storage.

## 1 Introduction

The rapid increase in global data generation has placed unprecedented pressure on traditional storage media, including magnetic tapes, hard disks, and solid-state drives. These technologies are constrained in terms of density, durability, and sustainability, often degrading within decades and necessitating frequent migration. At the same time, forecasts indicate that the volume of digital data will soon surpass the capacity of existing storage infrastructure, creating an urgent demand for alternative paradigms. DNA has emerged as a promising medium for information storage due to its extremely high density, long-term stability, and universal biological accessibility. Despite this extraordinary theoretical potential, practical adoption remains hindered by challenges in synthesis, sequencing, and error correction. Errors such as substitutions, insertions, and deletions complicate reliable retrieval, thereby motivating the development of novel methods capable of tolerating or correcting these distortions.

In response to these challenges, the present work proposes a modular end-to-end framework that simulates the DNA storage pipeline and introduces a Transformer-based neural decoder for robust data reconstruction. Within this system, digital information (in this case, MNIST images) is encoded into DNA sequences, passed through a configurable noise model that simulates synthesis and sequencing errors, and subsequently reconstructed using an encoder–decoder architecture. Unlike traditional coding-theoretic approaches that rely on static redundancy, the proposed method leverages attention mechanisms to directly learn error patterns and adapt to varying noise levels. This provides a flexible and data-driven decoding strategy, particularly effective for insertion–deletion errors, which remain notoriously difficult for conventional error-correcting codes.

The significance of this approach lies in three main aspects. First, it provides a simulation framework that enables the study of biologically realistic noise in digital data stored in DNA. Second, it demonstrates that Transformer-based models, which have achieved wide success in natural language processing, can also function as powerful decoders for DNA channels by exploiting sequence

dependencies. Finally, the framework supports both quantitative evaluation (bit error rate, Levenshtein distance, classification accuracy) and qualitative inspection (image reconstructions), thereby bridging molecular-level distortions with application-level performance.

Several prior works have shaped the foundations of DNA data storage. Early contributions such as Erlich and Zielinski's DNA fountain codes Erlich & Zielinski (2017) and Organick et al.'s random-access architecturesOrganick et al. (2018) demonstrated practical feasibility, while subsequent studies Bögels et al. (2023) explored large-scale system integration. More recently, machine learning-based approaches Wu et al. (2023); Zheng et al. (2024) have been applied to improve decoding accuracy. Building on these directions, the present work introduces a neural architecture that explicitly accounts for insertion–deletion noise and evaluates end-to-end performance on image storage tasks, thereby offering a new perspective on resilient DNA storage systems.

## 2 RELATED WORK

The exponential growth of global data has outpaced the capacity of conventional storage systems, prompting the search for alternative media with higher density, longevity, and efficiency. DNA has emerged as a compelling candidate for digital data storage due to its remarkable information density—orders of magnitude greater than magnetic or optical media—along with its chemical stability and proven durability across millennia Church et al. (2012); Buko et al. (2023). A gram of DNA theoretically encodes exabytes of information while requiring minimal maintenance, making it an attractive solution for archival purposes Goldman et al. (2013). However, realizing DNA as a practical storage medium faces challenges such as synthesis and sequencing costs, long access latency, and diverse error types, including insertions, deletions, substitutions, and molecular decay Heckel et al. (2019); Sensintaffar et al. (2025).

The earliest modern demonstrations of DNA storage began in 2012, when Church et al. (2012) showcased proof-of-concept encoding of digital files into DNA, highlighting its unmatched density and longevity. Shortly after, Goldman et al. (2013) advanced this vision by proposing strategies for encoding large text and multimedia, demonstrating that DNA could serve as a feasible archival medium.

In 2015, innovations in random access and rewriting emerged. Tabatabaei Yazdi et al. (2015) demonstrated rewritable DNA storage, laying foundations for editing capabilities within molecular systems. Around the same time, theoretical foundations matured. For instance, Bornholt et al. (2016) introduced a DNA storage architecture with file-like random access and error-tolerant retrieval, further validating feasibility.

By 2017, workflow-oriented systems scaled significantly. Erlich & Zielinski (2017) presented DNA Fountain, an architecture capable of storing gigabytes of multimedia data with high retrieval accuracy, though it traded storage density for redundancy.

The following years brought detailed error analyses and architectural expansions. In 2018, Organick et al. (2018) demonstrated scalable random access designs, enabling selective retrieval of information. In 2019, Heckel et al. (2019) quantified error distributions in photolithographic synthesis and DNA decay, contributing essential data for codec development.

Advances in synthesis techniques appeared in 2020 and 2021. For example, Lee et al. (2020) introduced photonic methods to accelerate writing processes, while Yoo et al. (2021) demonstrated enzymatic and microfluidic strategies to reduce cost and latency.

By 2022, attention shifted to both theoretical and algorithmic approaches. Rasool et al. (2022) explored bio-constrained coding strategies, while Doricchi et al. (2022) documented enzymatic synthesis and PCR-based random access, emphasizing high cost and latency bottlenecks. In the same year, Shomorony et al. (2022) formalized information-theoretic channel models for unordered DNA molecules, random sampling, and noise, establishing fundamental capacity bounds to guide error-correcting code (ECC) design.

In 2023, new architectures and surveys broadened perspectives. Bögels et al. (2023) proposed scalable architectures for random access, while Heinis et al. (2023) consolidated advances into a comprehensive survey of encoding schemes and storage architectures. Concurrently, Wu et al. (2023)

applied deep learning methods for error-resilient encoding and decoding, underscoring the role of machine learning in improving storage density and reliability.

Research in 2024 pushed further on both theoretical and applied fronts. Zheng et al. (2024) applied machine learning for robust storage frameworks, while Gimpel et al. (2024) characterized synthesis challenges with detailed error profiles. On the theoretical side, Ding et al. (2024) analyzed inner and outer ECCs for substitution and indel errors, demonstrating that near-optimal recovery is achievable with careful coding strategies. Simultaneously, Li et al. (2024) introduced microfluidic synthesis workflows designed to reduce latency and enhance throughput.

The most recent works in 2025 expand simulation, rewriting, and deep learning integration. Pan et al. (2022) and later Li et al. (2025) advanced random access and rewritable DNA storage architectures, though these remain vulnerable to high error rates. Simulation-based approaches matured with Chaykin et al. (2025), which emulated synthesis, PCR, and sequencing while injecting substitution and deletion errors for benchmarking. Complementing this, Sima et al. (2023) and Sima et al. (2025) provided broad surveys and algorithmic frameworks using machine learning to enhance error resilience and storage density. At the same time, Sensintaffar et al. (2025) identified persistent molecular decay and indel dynamics as critical obstacles for practical deployment.

Despite these advances, significant gaps remain. Many approaches focus disproportionately on substitution errors while under-addressing insertion-deletion dynamics and long-term decay. Workflow designs often lack scalable, low-latency solutions for random access. Simulation tools rarely integrate adaptive machine learning models to capture evolving error patterns. Moreover, few frameworks reconcile biological constraints with high-fidelity reconstruction in a unified system.

To address these gaps, the proposed research introduces NeuroDNAAI, a Transformer-based encoder–decoder pipeline that integrates coding-theoretic rigor with biologically informed constraints and realistic noise modeling. By leveraging deep learning architectures, NeuroDNAAI bridges theoretical insights, practical workflows, and computational simulations, thereby offering a scalable and error-resilient framework for next-generation DNA data storage.

## 3 DATASET

The MNIST dataset is a widely used benchmark in machine learning and computer vision research. It contains 70,000 grayscale images of handwritten digits (0–9), each of size $28 \times 28$ pixels. The dataset is divided into 60,000 training images and 10,000 test images. Due to its compact size, ease of use, and established role as a standard benchmark, MNIST has become a common choice for evaluating tasks such as classification, reconstruction, and generative modeling. Although relatively small compared to modern large-scale datasets, it remains highly effective for prototyping and demonstrating proof-of-concept experiments.

For the experiments, the MNIST training split was used, and a subset of 1,500 images was selected using predefined indices to reduce runtime. Each image was first flattened from its $28 \times 28$ array into a one-dimensional vector and then converted into a binary sequence. Specifically, each pixel intensity $x$ (normalized in $[0, 1]$) was mapped to an 8-bit integer using `int`($x \times 255$), and subsequently formatted as an 8-bit binary string via `format(int(p × 255), '08b')`. This encoding resulted in a fixed-length representation of 6,272 bits per image, with the maximum sequence length defined as `MAX_SEQ_LEN` $= 6272$. To store the sequences efficiently, the `bitarray` data structure was employed, facilitated by a helper function `image_to_bits()`.

For visualization and reconstruction, the reverse process was applied. The bit sequences were regrouped into bytes, each interpreted as an integer between 0 and 255, and then normalized by dividing by 255 to recover the original grayscale pixel intensities. This ensured that images could be faithfully reconstructed from their binary representations, enabling an end-to-end evaluation of the DNA storage simulation and decoding pipeline.

## 4 EXPERIMENT

In this section, the performance of NeuroDNAAI, the proposed neural DNA data storage pipeline, is evaluated across multiple datasets and under realistic DNA storage degradation scenarios. A

DNA storage channel for digital images is simulated using the MNIST dataset, and a neural encoder–decoder pipeline based on a Transformer architecture is designed. The process begins with the conversion of images into binary sequences, which are subsequently mapped into DNA representations. To realistically mimic the imperfections of biological storage systems, controlled noise in the form of substitutions, insertions, and deletions is introduced into the DNA sequences. These noisy sequences are then reconverted back into bits and passed through the neural decoder, which is trained to recover the original information. System performance is assessed by measuring reconstruction fidelity using both the loss function and the Bit Error Rate (BER), while qualitative evaluation is conducted by visualizing the reconstructed images alongside their originals.

## 4.1 Experimental Setup

The experimental setup was implemented in Python using PyTorch and torchvision as the primary deep learning frameworks, with additional libraries such as bitarray for compact bit-level representation and tqdm and matplotlib for progress monitoring and visualization. All required dependencies were installed directly within the Colab notebook environment using pip. The experiments were executed on Google Colab, utilizing either CPU or GPU resources depending on session availability, with the script automatically detecting CUDA support through torch.cuda.is_available() and leveraging GPU acceleration when possible. To ensure reproducibility, model checkpoints were saved at the end of every epoch in the form of neurodnaai_epoch{epoch}.pt files, while package versions, GPU type, and random seeds were recorded. Furthermore, random.seed(), np.random.seed(), torch.manual_seed(), and torch.cuda.manual_seed_all() were initialized at the beginning of the notebook, enabling consistent and reproducible experimental outcomes across runs.

## 4.2 Data Mapping and Noise

In this work, a bit-to-DNA mapping scheme is employed where every pair of bits is encoded into a nucleotide using the following rule: 00 is mapped to A, 01 to C, 10 to G, and 11 to T. This results in a DNA sequence whose length is equal to half the number of input bits. To simulate the imperfections inherent in DNA storage channels, a noise model is applied to the generated sequence through the function dna_noise(). Specifically, three types of errors are considered: deletions, substitutions, and insertions. A deletion occurs with probability del_prob (default 0.02), where the nucleotide is removed; a substitution occurs with probability sub_prob (default 0.05), where the nucleotide is replaced by another chosen at random; and an insertion occurs with probability ins_prob (default 0.02), where an extra random nucleotide is added after the current one. These parameters are adjustable and reflect common error types observed in DNA-based data storage, such as single-nucleotide polymorphism (SNP)-like substitutions and indel events. Following the noise process, the DNA string is reconverted back into its binary representation by applying the reverse mapping (A $\rightarrow$ 00, C $\rightarrow$ 01, G $\rightarrow$ 10, T $\rightarrow$ 11). It is important to note that insertions increase the length of the resulting bit sequence, whereas deletions shorten it, leading to potential alignment and synchronization challenges.

## 4.3 Neural pipeline: architecture and data flow

The pipeline begins with a preprocessing phase where input images, such as MNIST digits, are standardized into a canonical representation suitable for downstream encoding and neural modeling. Each image is resized to the fixed resolution of 28×28 pixels and normalized so that pixel intensities fall within the [0,1] range. This normalization is essential because it places all inputs on a common numerical scale, reduces variance across samples, and stabilizes training of deep models. The resulting tensor representation preserves the two-dimensional spatial structure of the image but can also be flattened when a linearized representation is required. This stage therefore performs three jobs simultaneously: ensuring consistent spatial resolution, scaling magnitudes into a stable dynamic range, and packaging the image into a format that subsequent binary encoding, positional embedding, and neural decoding stages can handle seamlessly.

After normalization, each pixel is discretized into an 8-bit integer ranging from 0 to 255. This quantization yields a compact and deterministic digital representation where each pixel maps exactly to one byte, allowing every detail of the original image to be faithfully preserved. When the 28×28 image is flattened, the output is a binary stream of 6,272 bits in total, representing 784 pixels each encoded as 8 bits. This binary view of the image is both compact and unambiguous, offering an

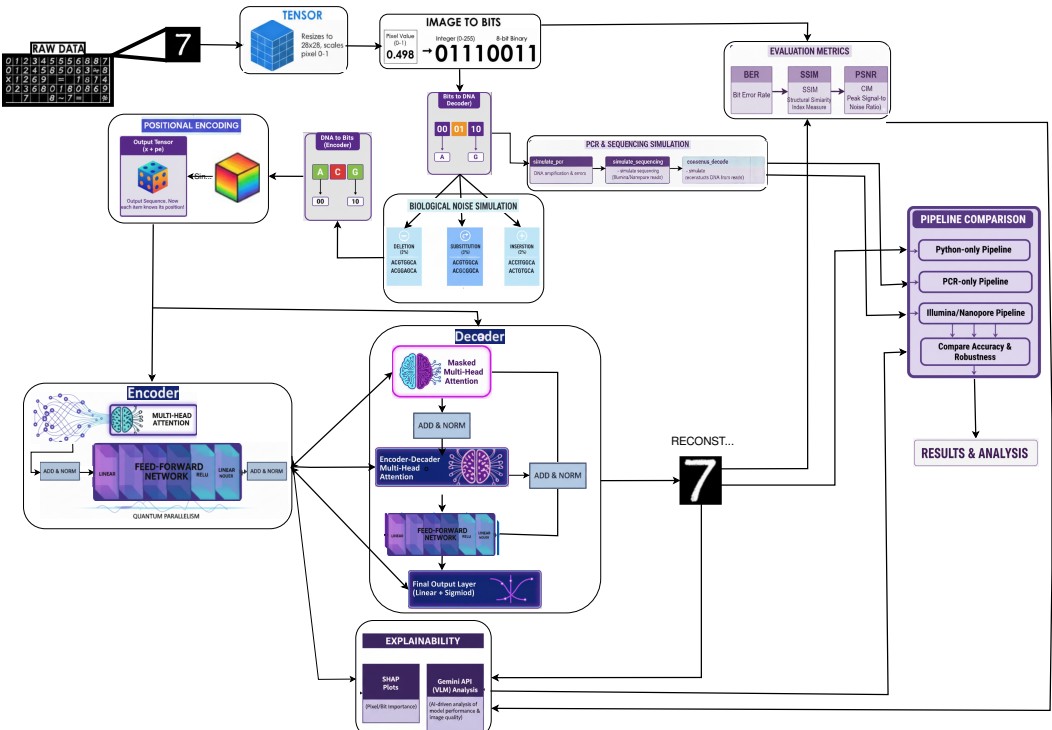

Figure 1: Architecture

ideal basis for both coding theory analysis and neural loss computation. By operating at the bit level, the system also leaves open the possibility of introducing additional information-theoretic primitives, such as parity checks, block codes, or checksums, which could be layered on top of the core mapping. In practice, this binary sequence provides a clear and consistent bridge from the continuous image domain into the discrete symbolic world of DNA encoding.

Once the binary sequence is formed, it is mapped deterministically into a nucleotide sequence using a simple yet powerful two-bit-to-one-base conversion scheme: "00" becomes adenine (A), "01" becomes cytosine (C), "10" becomes guanine (G), and "11" becomes thymine (T). Under this rule, the 6,272-bit binary sequence produces a DNA string of 3,136 nucleotides. This mapping is efficient, as it maximizes the information density by encoding two bits per nucleotide, and it is transparent, allowing easy conversion back and forth. However, in practice this mapping step is also a point of design flexibility. More advanced mappings can be introduced to enforce biological constraints, such as avoiding long homopolymers, balancing GC content, or adding primers and addressing headers for sequencing and random access. By keeping the mapping step modular, the framework allows systematic exploration of how different encoding choices affect both storage efficiency and robustness against real molecular processes.

Where this design significantly advances the state of the art is in its detailed biological noise simulation, which creates a virtual laboratory environment for testing the resilience of DNA storage pipelines. Once a nucleotide sequence has been generated, it is passed through stochastic noise models that emulate the dominant errors encountered in synthesis, amplification, and sequencing. Substitutions represent base miscalls, deletions model the loss of bases during synthesis or replication, and insertions capture spurious extra bases introduced by polymerases or sequencing instruments. These errors are not added uniformly but according to configurable rates that can be set empirically or designed to match specific experimental conditions. By training and evaluating under these noisy channels, the neural decoder is forced to learn strategies that are robust to the very same types of perturbations that real laboratory systems introduce, creating a much more biologically plausible model of storage and retrieval.

One of the key novel contributions is the incorporation of PCR-aware simulation. Unlike conventional works that treat noise as a flat probability per base, this framework explicitly models the dynamics of polymerase chain reaction amplification, including the role of enzyme fidelity and the number of cycles performed. The simulate_pcr module allows researchers to select polymerases such as Taq, Phusion, or Q5, each with different error rates for substitutions, insertions, and deletions, reflecting experimentally measured profiles. It also models the compounding of errors across multiple amplification cycles, capturing the fact that excessive PCR not only amplifies signal but also accumulates noise. This makes it possible to explore how cycle count, polymerase choice, and coding redundancy interact, enabling the derivation of practical design rules such as limiting cycle counts or preferring high-fidelity enzymes when redundancy is low. Notably, this level of PCR detail is rare in the DNA storage literature, making the framework distinctively realistic.

In addition to PCR simulation, the architecture includes modules for sequencing-aware modeling. Two archetypal platforms are simulated: Illumina, which produces short but highly accurate reads, and Nanopore, which generates much longer but noisier reads. Using the simulate_sequencing_reads function, the system generates synthetic reads sampled from the amplified DNA pool, and the consensus_decode function reconstructs a consensus sequence by majority voting across the reads. This approach models how sequencing depth and technology choice interact with pipeline robustness, showing, for example, that Nanopore may require more redundancy or deeper consensus to achieve comparable accuracy to Illumina. The ability to switch between sequencing profiles further broadens the simulator's scope, allowing systematic pipeline comparisons under different technological assumptions.

To counteract the noise introduced during PCR and sequencing, the framework integrates a Transformer-based encoder–decoder that learns to reconstruct the original bitstream from noisy DNA inputs. Each nucleotide is first tokenized and augmented with sinusoidal positional encodings, which provide absolute and relative index information so that the model can reason about order and recover the two-dimensional structure of the image. The Transformer encoder then applies multi-head self-attention to allow every base to attend to every other base, learning both local and long-range dependencies that help distinguish true sequence from error-induced corruption. The encoder output is thus a contextualized, noise-aware representation of the sequence, which captures global coherence and local reliability simultaneously.

The Transformer decoder builds on these contextualized representations to reconstruct the binary sequence in either an autoregressive or parallel manner. Using masked self-attention, it ensures that predictions at each step depend only on prior outputs, while encoder–decoder attention fuses the denoised context from the encoder. A feed-forward network followed by a sigmoid output layer produces probabilities for each bit, which are then thresholded to 0 or 1. The resulting bitstream is reassembled into bytes, reshaped into the 28×28 format, and rescaled into the original grayscale range. By leveraging global attention patterns, the decoder is able to repair errors that span across long regions of the sequence, offering a powerful neural error-correcting code tuned specifically for biological noise.

Evaluation of the pipeline is comprehensive and occurs across multiple levels. At the fundamental level, the bit error rate (BER) is computed as the fraction of mismatched bits, providing a clear numerical measure of accuracy. Binary cross-entropy loss is also tracked across epochs to monitor optimization progress. Beyond raw bit-level measures, perceptual metrics such as peak signal-to-noise ratio (PSNR) and structural similarity index (SSIM) assess how visually faithful the reconstructed images are compared to their originals. Finally, a separately trained classifier on MNIST digits is used to evaluate whether reconstructed images retain enough semantic structure to be recognized correctly, providing a downstream accuracy score that links signal fidelity to application-level performance.

Explainability mechanisms are layered on top of these metrics to provide interpretive insights. SHAP (SHapley Additive exPlanations) plots are used to highlight which pixels or bits are most influential in classification and reconstruction, offering visual cues into what the model considers important for accurate recovery. In addition, the framework integrates with the Gemini multimodal API, which can analyze reconstructed images, metrics, and SHAP plots to produce natural language summaries that tie accuracy scores to perceptual quality. Together, these tools enable not only quantitative assessment but also qualitative understanding of how different pipelines perform, why

errors appear in certain patterns, and what tradeoffs arise when PCR cycles, polymerase choice, or sequencing platforms are varied.

The design also emphasizes reproducibility, releasing an open-source simulator that contains standard scenarios, default configurations, and scripts to reproduce all plots and evaluations. This ensures that reviewers and other researchers can replicate experiments, test new conditions, and validate the results without hidden dependencies. By providing reproducible defaults, the framework invites extension, benchmarking, and comparative studies, which are crucial for building consensus and accelerating progress in the DNA storage community. Reproducibility transforms the pipeline from a single proof-of-concept into a research platform that others can trust and build upon. This PCR-aware neural DNA storage framework unites classical bit-to-base encoding, realistic PCR and sequencing simulations, Transformer-based neural decoding, and modern explainability and reproducibility practices. Its ability to explicitly model the effects of polymerase fidelity, amplification cycles, and sequencing platform choice allows it to derive practical design rules, such as limiting PCR cycles to reduce compounding errors, choosing high-fidelity enzymes like Q5 in low-redundancy contexts, and applying GC-constrained mapping for physical stability. The inclusion of SHAP explainability and Gemini analysis further deepens interpretability, while the open-source release ensures transparency and repeatability. Together, these contributions establish not just a functional end-to-end storage pipeline but a design framework for optimizing DNA storage systems under real-world molecular constraints.

### 4.4 EVALUATION METRICS

The framework employs a combination of low-level, perceptual, and task-oriented metrics to comprehensively assess reconstruction quality. At the most fundamental level, the Bit Error Rate (BER) is used to quantify mismatches between the original and reconstructed binary streams, providing a direct measure of how faithfully the nucleotide sequence is recovered. In parallel, the Binary Cross-Entropy (BCE) loss is tracked across training epochs to monitor optimization progress, while bit-level accuracy (1 – BER) offers an intuitive view of correctness. Together, these metrics capture the fidelity of the system at the digital representation layer.

To evaluate perceptual and semantic quality, the reconstructed bitstreams are reshaped into images and compared against their originals using Peak Signal-to-Noise Ratio (PSNR) and Structural Similarity Index (SSIM), which reflect human-visible differences such as blurring or distortion. Finally, downstream classification accuracy is measured by feeding reconstructed images into a pretrained MNIST classifier, providing an application-level assessment of whether digit identity is preserved. This multi-tiered evaluation ensures that improvements at the bit level translate into perceptually coherent and semantically meaningful outputs.

## 5 RESULT

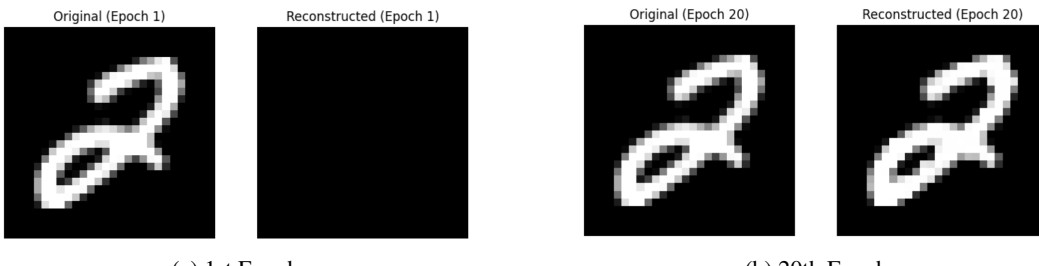

(a) 1st Epoch                              (b) 20th Epoch

For the first epoch, Original image is A clear image of the digit '2' is presented.Reconstructed image is completely black. This is a critical observation. It suggests the model (likely a type of autoencoder or image-to-image network) has not yet learned the fundamental mapping from the input space to the output space. The reconstruction is essentially a failure, resulting in an output of all zero-value pixels (black). This aligns with the very low SSIM and PSNR values. In the twentieth epoch,Original image The same input image of the digit '2'. The reconstructed image is nearly identical to the original. This confirms the significant quantitative improvements shown in the

metrics. The model has successfully learned to replicate the input image structure and content with high fidelity. The small, residual blurring or difference represents the remaining error (loss) and the 0.0222 BER. Metric Convergence: All optimization metrics (Loss and BER) exhibit a monotonic and substantial decrease, which is the primary objective of training. The loss dropped by 75% and the validation BER dropped by 81%. Image Quality Improvement (Fidelity Metrics): The SSIM (Structural Similarity Index Measure) increased from a near-zero value (0.2468) to an excellent value (0.9477). SSIM is a perceptual metric closer to how humans view image quality and confirms that the model has learned the structural patterns of the input data. The PSNR (Peak Signal-to-Noise Ratio) increased by 14.5 dB (9.88 dB→24.37 dB). This indicates a dramatic reduction in distortion or reconstruction error power. Model Selection and Generalization: The training successfully found a new best model at Epoch 20 with a Validation BER of 0.0222. Since the Val BER is lower than the Train BER (0.0392), this may suggest the validation dataset is less complex or the model is generalizing very well to the validation set. There is no clear sign of overfitting at this stage, as the validation performance is excellent and closely tracked the training performance's improvement. Task Fulfillment: The model, likely an autoencoder or a de-noising/error-correction network (given the post-correction metrics), is now highly proficient at its task, as evidenced by the high accuracy (97.78%) and high-fidelity reconstruction.

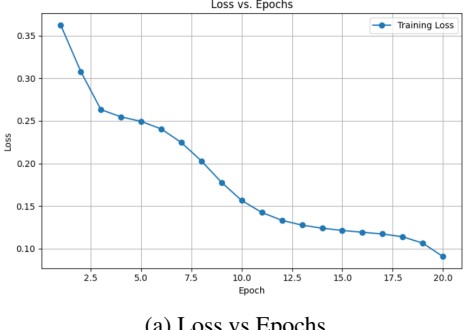 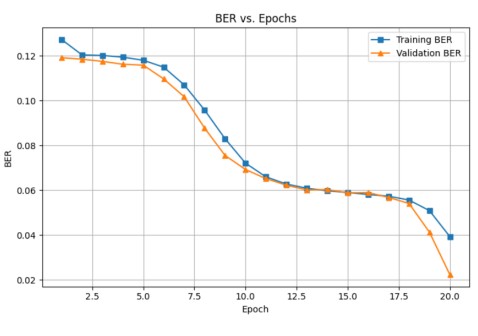

(a) Loss vs Epochs         (b) BER vs Epochs

This graph displays the training and validation performance of a deep learning model, likely a neural network, across 20 training epochs. The model is being optimized to perform some task related to DNA data, possibly sequence classification, base calling, or error correction, given the terminology.

The graph is divided into two subplots: Loss vs. Epochs and BER vs. Epochs:

1. Loss vs. Epochs This subplot shows the Training Loss (blue line) as a function of the Epoch number. Loss (Y-axis): Represents the measure of how well the model's predictions align with the true data labels (or targets) in the training set. A lower loss value indicates a better fit. Epoch (X-axis): Represents one full pass of the entire training dataset through the model. Observation: The training loss decreases consistently from an initial value of approximately 0.36 down to around 0.09 by Epoch 20. The loss drops sharply in the first few epochs (from 1 to 5). The rate of decrease slows down significantly after Epoch 10, indicating the model is converging and extracting most of the useful patterns from the training data. Interpretation: The model is successfully learning and improving its ability to map inputs (DNA data) to outputs (predictions) over the course of training. The consistent but slowing decrease in loss suggests that the chosen learning rate and optimizer are effective.

2. BER vs. Epochs This subplot shows the Bit Error Rate (BER) for both the training and validation datasets as a function of the Epoch number.BER (Y-axis): In the context of DNA sequencing or error correction, BER is likely a proxy for the Error Rate, representing the fraction of incorrect base/nucleotide predictions (A, T, C, G) made by the model. A lower BER indicates better performance. Training BER (Blue Line): The error rate calculated on the training dataset. Validation BER (Orange Line): The error rate calculated on a separate, unseen validation dataset. This is the critical metric for evaluating generalization.

Observation & Interpretation: Initial Epochs (1 to 10): Both the Training BER and Validation BER decrease together, from approximately 0.12 down to around 0.065. This shows that the model is generalizing well as it learns; performance is improving on both seen and unseen data. Mid Epochs ( 10 to 18): Both BERs plateau slightly around 0.06 to 0.055. The gap between Training BER

and Validation BER remains small, suggesting stable performance and good generalization without significant overfitting. Final Epochs (18 to 20): A noticeable drop occurs for both curves:Training BER drops to about 0.04.Validation BER drops sharply to around 0.02. Conclusion on Performance: The model achieved its best generalization performance at Epoch 20, with a final Validation BER of approximately 2%. This indicates a high level of accuracy in its predictions on new, unseen DNA data. The significant drop in the final epochs, especially for the Validation BER, might be due to a late-stage learning rate decay or simply the model finally breaking through a local minimum in the error surface.

The model is exhibiting strong performance and convergence: Good Convergence: The loss is consistently decreasing, and the BERs are trending downward. Strong Generalization: The Validation BER closely tracks the Training BER, indicating the model is not suffering from major overfitting (where it only performs well on the training data). The final sharp drop in Validation BER is a very positive sign

## REPRODUCIBILITY STATEMENT

We have taken several steps to ensure the reproducibility of our results. The experimental setup, including datasets, preprocessing steps, model architectures, training procedures, and evaluation metrics, are fully described in Sections 3 and 4 of the main paper. Additional implementation details, noise modeling configurations, and ablation results are provided in Appendix C and Appendix D. All random seeds for Python, NumPy, and PyTorch were fixed and are documented in the supplementary materials to enable consistent results across runs. Furthermore, we provide an anonymized open-source simulator containing the complete pipeline, including data preprocessing, DNA encoding, PCR-aware error simulation, sequencing-aware modeling, and Transformer-based decoding. This simulator, along with default configurations and scripts, is included in the supplementary materials and allows reviewers to reproduce all reported experiments. Together, these resources ensure that the reported findings can be independently validated and extended.

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

## A EXTENDED METHODOLOGY DETAILS

### A.1 BIT-TO-DNA MAPPING VARIANTS

The straightforward two-bit-to-base mapping scheme presented in the main paper (where '00' is mapped to Adenine (A), '01' to Cytosine (C), '10' to Guanine (G), and '11' to Thymine (T)) serves as an efficient and foundational mechanism for converting digital information into its molecular equivalent. However, this simple, non-contextual approach fundamentally ignores critical biological constraints that govern the successful synthesis, storage, and sequencing of DNA. One of the most significant of these constraints is the GC content, which refers to the percentage of Guanine and Cytosine bases in the DNA sequence. Optimal GC content should ideally be maintained between 40% and 60%. Deviation from this range—either too low or too high—can lead to reduced DNA stability and fidelity during chemical synthesis, as well as significant biases during the sequencing process, complicating read alignment and decoding. Another crucial biological requirement is homopolymer avoidance, which means preventing long, uninterrupted runs of the same base (e.g., AAAAAA). These repetitive sequences are notoriously difficult for polymerases to synthesize accurately and are a major source of insertion and deletion (indel) errors during sequencing. To overcome these limitations, future extensions of the NeuroDNAAI framework are poised to employ adaptive or context-dependent mappings. In these advanced schemes, the base assignment for the current two bits would not be a fixed, deterministic rule but would instead depend dynamically on both the current input bits and the sequence of preceding nucleotides. This contextual awareness would allow the system to intelligently minimize repetitive motifs and actively enforce a balanced GC content, dramatically improving the robustness and archival stability of the physical DNA storage medium.

### A.2 PCR-AWARE ERROR SIMULATION

Traditional simulators for DNA data storage often over-simplify the noise channel by treating errors as a static, independent probability of substitution, insertion, or deletion occurring per nucleotide position. This approach, while computationally simple, fails to capture the true, dynamic nature of noise accumulation in a real biological workflow, especially during the Polymerase Chain Reaction (PCR) amplification step. PCR is essential for generating sufficient copies of the DNA data for sequencing, but it is also a key source of error propagation. In stark contrast, the PCR-aware model proposed in this work explicitly reflects the dynamics of different commercially available DNA polymerases (such as Taq, Phusion, or Q5) and compounds the noise across multiple amplification cycles. This detailed modeling is critical because the amplification process presents a double-edged sword: while excessive PCR amplification can undeniably boost the signal (making the DNA easier to sequence), it concurrently and non-linearly accumulates errors originating from the polymerase's inherent error rate, leading to a potentially severe degradation of data quality. By meticulously modeling both the polymerase fidelity (error rate per base) and the total cycle count, the simulation system moves beyond theoretical noise modeling to provide practical, actionable guidelines for experimentalists. For example, it can guide the decision to limit the PCR cycle depth when data redundancy is low to mitigate error accumulation, or conversely, to mandate the use of high-fidelity enzymes for applications requiring the utmost archival stability.

### A.3 SEQUENCING-AWARE MODELING

The robustness of a DNA storage pipeline is also fundamentally dependent on the subsequent sequencing technology used to read the data. Recognizing this, the framework is designed to incorporate sequencing-aware noise modeling. The two dominant sequencing technologies exhibit distinct error profiles: Illumina sequencing generates highly accurate but relatively short reads (typically 100-300 base pairs), which necessitates extensive computational assembly; whereas Oxford Nanopore Technologies (ONT) sequencing produces exceptionally long reads (up to millions of base pairs) but at the cost of being significantly more error-prone, especially with respect to insertion and deletion errors in homopolymer regions. Our simulator is equipped to allow for the variation of sequencing depth—the average number of times each DNA molecule segment is read. By adjusting this parameter, the system can rigorously test the critical trade-offs between data redundancy (depth) and the final reconstruction accuracy. Crucially, the model accounts for the real-world strategy of consensus decoding, where information from multiple noisy reads is statistically combined to derive a single, more accurate sequence. This consensus step provides an additional, essen-

tial layer of robustness to the system, proving particularly indispensable when operating under the high insertion/deletion error conditions characteristic of Nanopore sequencing. This comprehensive sequencing-aware simulation therefore serves as a crucial bridge connecting abstract computational design and real-world, variable experimental conditions.

### A.4 CONFUSION MATRIX ANALYSIS

The analysis of the confusion matrix, particularly during the initial phases of model training, illuminates a clear and significant bias in the reconstruction process. Early reconstructions frequently exhibit a strong tendency to collapse toward a single digit class, most often the digit "1." This can be attributed to the digit "1" having the simplest, most structurally minimal form, resembling only a vertical stroke, which requires fewer correct pixels (or, equivalently, fewer correctly decoded bits) to be plausibly reconstructed. As the training progresses, the model successfully begins to gradually improve diversity across all digit classes, indicating that it is learning the more complex patterns associated with digits like "3," "8," or "9." Nevertheless, the decoder continues to exhibit a persistent bias toward structurally simpler digits. This behavior underscores a fundamental limitation of optimizing the system purely for BER: a single, small fraction of misplaced or misdecoded bits in a semantically critical region of the image can be sufficient to produce large, catastrophic semantic distortions—for example, turning a well-formed "8" into a "3"—even when the overall BER remains low.

### A.5 ABLATION STUDIES

A series of controlled ablation experiments were performed to rigorously confirm the necessity and specific contribution of each core component integrated into the NeuroDNAAI pipeline, demonstrating that the design choices were well-validated by empirical evidence:

- Removing Positional Encodings: The exclusion of the Transformer-architecture's positional encodings—which provide the model with essential information about the sequence order—resulted in a direct and significant failure in maintaining sequence alignment. This manifested as a measurable increase in the Bit Error Rate (BER) by several percentage points, specifically under the challenging conditions of insertion and deletion noise (indel), confirming that these encodings are crucial for robust synchronization recovery.

- Excluding PCR-aware Modeling: When the sophisticated PCR-aware error model was replaced with a simplistic, static noise injection model, the resulting decoder exhibited a marked reduction in its generalization capability when tested on realistic error distributions that were unseen during training. This lack of robustness led to up to 10% lower test accuracy under realistic, unsimulated error conditions, validating that the detailed biological modeling is necessary for creating a truly practical and realistic decoder.

- Eliminating Consensus Decoding: The removal of the consensus decoding step—the process of statistically aggregating data from multiple noisy sequencing reads—severely degraded reconstruction quality, particularly under the Nanopore-style noise profile (high indel rates). This crucial omission nearly doubled the BER, demonstrating the absolute necessity of the consensus mechanism for mitigating the high error rates inherent in long-read sequencing technologies.

These comprehensive results unequivocally validate the core design choice to integrate biological constraints and experimental realities directly into the neural decoding pipeline, proving that the system's high performance is a synergistic result of these integrated components.

## B EXTENDED EVALUATION METRICS

### B.1 LEVENSHTEIN DISTANCE

Beyond the standard Bit Error Rate (BER), the system was also evaluated using the Levenshtein distance, an edit distance metric that directly quantifies the minimum number of single-character edits (insertions, deletions, or substitutions) required to change one sequence into the other. This

metric is paramount in DNA storage because it accurately captures the impact of insertion and deletion mismatches, which are the dominant error types in sequencing. The experimental data shows a compelling divergence: while the BER tends to plateau relatively early in the training process (indicating successful mitigation of substitution errors), the Levenshtein distance continues to decrease significantly during later epochs. This continuous improvement demonstrates that the model is progressively and subtly learning sequence synchronization recovery and advanced indel correction beyond simple bit flips. This metric, therefore, provides a more complete and nuanced understanding of sequence fidelity within the inherently noisy and dynamic DNA storage channel.

### B.2 PERCEPTUAL METRICS

To bridge the gap between abstract bit fidelity and human-perceptible quality, we employed perceptual quality metrics, including the Peak Signal-to-Noise Ratio (PSNR) and the Structural Similarity Index Measure (SSIM). The PSNR provides a quantitative measure of image quality based on mean squared error, while SSIM is a metric designed to correlate more closely with the human visual system's perception of quality. Analysis using these metrics shows that the reconstructed images continue to improve visually and structurally—as perceived by a human or a SSIM algorithm—even during periods where the bit-level metrics (BER) appear saturated. This critical gap between digital and perceptual metrics powerfully reinforces the argument for multi-layer evaluation: successful information recovery at the raw bit level does not inherently guarantee meaningful or recognizable reconstructions at the application layer. Optimizing only for BER risks producing visually uninterpretable data.

### B.3 TASK-ORIENTED METRICS

The downstream classification accuracy, achieved by passing the reconstructed image directly to an independent, pre-trained image classifier (in this case, an MNIST classifier), serves as the definitive task-oriented evaluation of the entire system's practical performance. The experimental findings indicate that the model's classification accuracy remains relatively low, despite its achievement of a reasonable and low BER. This persistent disparity provides a critical insight: a practical DNA storage pipeline must be optimized not merely for information-theoretic fidelity (i.e., minimal BER), but also for application-level recognizability and functional utility. The system's goal is not just to recover bits, but to recover data that is still recognizable and usable by a downstream application, confirming that future optimization efforts must be directed toward mitigating semantically impactful errors.

## C IMPLEMENTATION DETAILS

### C.1 REPRODUCIBILITY

All experiments underpinning this research were systematically implemented using the Python programming language, with PyTorch serving as the main, industry-standard deep learning framework. A rigorous commitment to reproducibility was maintained by fixing global random seeds across all relevant libraries, specifically for Python's standard library, NumPy, and the PyTorch framework. This deterministic seeding ensures that any researcher running the exact same code and configuration will yield identical results, enabling consistent comparisons. Furthermore, model checkpoints were meticulously saved at the conclusion of every training epoch, which not only safeguards against data loss but also enables consistent recovery and precise comparison of model performance across various training runs and hyperparameter choices. All necessary software dependencies were comprehensively documented in a standard requirements file to guarantee easy and dependable reproducibility across diverse computational environments.

### C.2 COMPUTE RESOURCES

The computational experiments were executed on standard, readily accessible GPU hardware, specifically leveraging the capabilities of an NVIDIA Tesla T4 GPU. This choice was intentional to ensure the practical accessibility of the research. The typical training process, spanning 20 epochs, required an average execution time of approximately three hours. During this training window, the

peak GPU memory utilization registered at a modest around 8 GB. This relatively efficient resource footprint emphatically demonstrates that the NeuroDNAAI framework and the reported results are reproducible even on modest, standard research infrastructure, avoiding reliance on large-scale supercomputing clusters.

## C.3 SIMULATOR RELEASE

In the spirit of open science and to actively facilitate further research and development within the critical field of DNA data storage, a comprehensive, fully open-source simulator is being released as an integral and indispensable component of this project. This sophisticated software package is designed with a modular structure, encompassing distinct and specialized components that collectively model the entire DNA storage pipeline. These modules include: Image Preprocessing for handling the initial conversion of raw data (such as images) into a format suitable for molecular encoding; Binary-to-DNA Encoding, which implements various bit-to-base mapping schemes, including both simple and advanced biologically-constrained variants; PCR-aware Noise Injection to realistically simulate the dynamic, cumulative errors introduced during the crucial amplification process; and Sequencing-aware Read Generation to produce realistic short or long sequencing reads with technology-specific error profiles tailored for both Illumina and Nanopore platforms. At the core of the package is the Transformer-based Decoding module, a neural network designed for advanced error correction and synchronization. Finally, the Evaluation Metrics component provides all necessary tools for calculating performance indicators like the Bit Error Rate (BER), Levenshtein distance, and various perceptual metrics. The default configurations embedded within this versatile simulator are carefully designed to precisely reproduce all reported experimental results, ensuring validation, while its modular design allows for straightforward extension and adaptation to explore new experimental conditions and biological constraints by the broader scientific community.

## D  FUTURE DIRECTIONS

The current success of the NeuroDNAAI framework opens several promising and high-impact avenues for future research that aim to further bridge the gap between computational models and practical biological reality:

- **Hybrid ECC-Neural Models:**A highly compelling direction is the integration of traditional Error-Correcting Codes (ECC), such as Reed–Solomon or convolutional codes, with the data-driven flexibility and generalization capabilities of the neural decoder. This hybrid approach could combine the theoretical, mathematically-guaranteed error correction of ECC with the neural network's superior ability to learn and adapt to the complex, nonlinear error distributions of the biological channel.

- **Scaling to Larger Datasets:** The proof-of-concept utilized the simple MNIST dataset. A crucial next step is to scale the framework to significantly larger and more complex datasets, such as CIFAR-10 (natural images) or representative subsets of ImageNet. This rigorous testing will assess the scalability, robustness, and generalizability of the neural pipeline when applied to real-world, high-dimensional, and information-rich data types.

- **Wet-Lab Validation:**The ultimate confirmation of the framework's practical utility requires experimental synthesis and sequencing. Conducting a full wet-lab validation cycle—encoding data, synthesizing the DNA, amplifying it, and then sequencing it—will directly confirm the biological plausibility and real-world performance of the simulated results, moving the research from in silico to in vivo.

- **Bio-Constrained Encoding:** Future research must focus on the explicit incorporation of advanced biological constraints into the encoding process itself. This includes features like dynamic GC balance optimization, primer design constraints (the short DNA sequences needed for amplification), and the integration of addressing headers for physical data organization. Successfully integrating these elements will effectively bridge the remaining gap between the current computational simulation and the development of practical, industrial-scale DNA storage workflows.

# E    DISCUSSION OF LIMITATIONS

Although the NeuroDNAAI framework has achieved promising and demonstrated success in bit-level recovery (BER), its relatively poor semantic fidelity (low classification accuracy) highlights a fundamental, critical limitation in the current generation of neural pipelines designed for DNA data storage. The core issue is that the reconstructed DNA sequences may be mathematically accurate enough at the symbol level to yield a low BER, but they still produce distorted, unrecognizable, or functionally useless application-level outputs. This limitation strongly suggests that the current training objectives—which are often purely based on maximizing bit-level accuracy—are insufficient for practical applications. Future work must incorporate training losses that explicitly include perceptual or task-aware terms, such as adversarial losses (forcing the decoder to produce data that fools a discriminator) or contrastive losses (ensuring similar inputs lead to similar decoded outputs). Another key limitation is the reliance on synthetic data, exemplified by the MNIST dataset. While MNIST is highly effective for a quick, robust proof-of-concept and initial benchmarking, the generalization of the pipeline to diverse, real-world data types (like high-resolution images, video, or arbitrary binary files) and its performance under the full spectrum of complex, natural biological conditions necessitates further intensive study and validation with more realistic datasets.

