# OpenReview forum: "NeuroDNAAI: Neural Pipeline Approaches for Advancing DNA-Based Information Storage as a Sustainable Digital Medium Using Deep Learning Framework"
_ICLR.cc/2026/Conference — Submitted to ICLR 2026_

### Official Review · Reviewer_Wp8Z · 2025-10-20

**Soundness:** 1
**Presentation:** 2
**Contribution:** 2
**Rating:** 2
**Confidence:** 5

**Summary:**

This paper presents a Transformer-based decoder designed to correct errors inherent to DNA-based information storage encoded by two-bit-to-one-base mapping. The authors validated their approach with experiments on MNIST images. The authors have an research on end-to-end simulation of the DNA storage channel, including PCR and sequencing-aware noise models.

**Strengths:**

Applying deep learning techniques to bioinformatics is a promising field of research. Specifically, DNA-based storage shows significant potential as a future technology.

**Weaknesses:**

While the proposed framework is comprehensive, its novelty could be further clarified. The core concepts, including high-level DNA storage simulation and deep learning-based decoders that learn channel characteristics, have been explored in previous studies (see references below).

1. Hamoum, Belaid, et al. "Channel model with memory for DNA data storage with nanopore sequencing." 2021 11th International Symposium on Topics in Coding (ISTC). IEEE, 2021.
2. Yuan, Lekang, et al. "DeSP: a systematic DNA storage error simulation pipeline." BMC bioinformatics 23.1 (2022): 185.
3. Chaykin, Gadi, et al. "DNA-storalator: a computational simulator for DNA data storage." BMC bioinformatics 26.1 (2025): 204.
4. Guo, Alan JX, et al. "Disturbance-based Discretization, Differentiable IDS Channel, and an IDS-Correcting Code for DNA Storage." (2025).

## Justification
From the reviewer's perspective, the proposed method and its evaluation are not sufficiently justified.

+ The Transformer-based decoder is used to recover information from noisy sequences retrieved from the DNA storage pipeline. However, the encoder that transforms binary information into a DNA sequence does not add any redundancy for error correction. This approach appears to contradict fundamental principles of information theory.
+ The authors propose using a single DNA strand of 3,136 nucleotides as the storage medium. However, synthesizing a single strand of this length is not feasible with current biochemical technologies, which are typically limited to synthesizing sequences of up to approximately 200 nucleotides directly.
+ The experiments on MNIST are not convincing due to the dataset's inherent properties. MNIST images contain significant redundancy, especially with large areas of black pixels, which can make the decoder's performance appear better than it is. Additionally, these simple images are not sensitive enough to synchronization errors (indels), a key challenge in DNA storage, making MNIST a poor choice for rigorously evaluating the system's capabilities.
+ A convincing way for the authors to verify the effectiveness of their approach would be to encode a randomly generated binary string, recover it through the proposed method, and present the resulting BER.

**Questions:**

+ The questions raised in the Weakness section.
+ Line 018. The abstract states that the framework 'draws from quantum parallelism concepts' to enhance encoding diversity. Could the authors please clarify which specific concepts are being referenced and explain how they were applied to the framework?
+ Section 3 Dataset. It is unconventional to dedicate this much space in the main text to describing a well-known dataset like MNIST. This information is typically shortened or placed in an appendix.
+ The authors claim a “detailed biological noise simulation” as a core contribution; however, the experiments section provides little evidence or analysis to support this claim.
+ Section 4.3. Section 4.3 is difficult to read due to its length. Spanning four pages, this single subsection covers too many topics at once, which can confuse readers.

---

### Official Review · Reviewer_VFcw · 2025-10-28

**Soundness:** 1
**Presentation:** 1
**Contribution:** 1
**Rating:** 0
**Confidence:** 4

**Summary:**

The paper proposes NeuroDNAAI, a deep-learning–based framework for reconstructing images stored in DNA.

The system includes: (i) an image-to-bits-to-DNA mapping, (ii) a simulator that injects biological channel noise (substitutions, insertions, deletions) derived from real-life wet-lab processes like PCR or different sequencing technologies, and (iii) a transformer-based decoder intended to recover the original signal.

The authors address an important bottleneck in DNA storage, and employing a learned end-to-end framework for error correction with a realistic simulation of error patterns is an interesting idea.
However, the manuscript in its current form lacks clarity, validation and evidence for central claims.

All in all, while the idea seems interesting, the manuscript currently contains unsubstantiated claims, lacks experimental data, as well as comparison to state-of-the-art methods.

**Strengths:**

None.

**Weaknesses:**

- The abstract states that "traditional prompting or rule-based schemes" fail and that NeuroDNAAI achieves superior accuracy, plus "low bit error rates for both text and images". However, none of these results are shown, quantified, or referenced in the main text.
The manuscript does not provide concrete numbers or comparisons with state-of-the-art tools.

- "Quantum parallelism" appears without substance. It is referenced in the abstract and Figure 1 without any further mention in the text.

- Questionable wording in the text: In multiple locations in the text, speculative wording is used, e.g., "It suggests the model (likely a type of autoencoder or image-to-image network)" or "The model, likely an autoencoder or a de-noising/error-correction network (given
the post-correction metrics)".

- A lot of the text in the results section is allocated to trivial descriptions of the loss curve during training. Deeper, more interesting topics like the actual data representation or error coding learned by the network are not touched.

- No validation of the channel noise simulator is given. I would expect at least a reference to a source where the error statistics for different biological processes like PCR are explained.

- Figure 1 seems unpolished. The arrows are skewed, the arrow in the top right does not touch its box, and "Decoder" in the middle of the figure has a character weirdly overpainted. The heading "Reconst..." seems truncated. Some parts of the figure don't seem reasonable to me: (a) Why is the PCR & Sequencing simulation distinct from the biological noise simulator and the result not passed into the encoder? (b) What is "Python-only" pipeline? This does not seem semantically like an alternative to PCR or Illumina/Nanopore.

**Questions:**

None.

---

### Official Review · Reviewer_WUe9 · 2025-10-30

**Soundness:** 2
**Presentation:** 2
**Contribution:** 2
**Rating:** 0
**Confidence:** 4

**Summary:**

This paper introduces NeuroDNAAI, an end-to-end neural framework for DNA-based data storage that integrates biologically realistic simulations, PCR-aware noise modeling, and Transformer-based encoding and decoding. The method simulates the entire molecular workflow, from bit-to-base encoding and amplification noise to sequencing-aware reconstruction. It uses a Transformer decoder to recover data from noisy DNA channels. Experiments on the MNIST dataset show that NeuroDNAAI achieves low bit error rates and high reconstruction fidelity. The paper also provides open-source simulation tools, interpretability via SHAP analysis, and discusses extensions toward hybrid ECC-neural models and wet-lab validation.

**Strengths:**

1. The paper presents a comprehensive and biologically grounded simulation framework, covering encoding, PCR, sequencing, and neural decoding, which distinguishes it from most prior purely computational DNA storage works.

2. Its integration of PCR-aware and sequencing-aware modeling (e.g., enzyme fidelity, platform-specific errors) significantly enhances realism and practical relevance.

3. The Transformer-based decoder effectively learns to correct substitution, insertion, and deletion errors, achieving notable improvements in bit-level accuracy and perceptual reconstruction quality.

**Weaknesses:**

1. The experiments only rely on MNIST, a simple dataset that does not fully represent real DNA storage scenarios.

2. Evaluations can be conducted on larger or more diverse datasets.

3. Comparative analysis with prior neural approaches (e.g., Wu et al., 2023; Zheng et al., 2024) is missing, limiting quantitative benchmarking.

4. The entire study remains simulation-only; even partial wet-lab validation (e.g., synthesis of short oligos) would greatly enhance credibility.

5. The architecture in Fig. 1 is very blurry and not so clear.

6. The comparative analysis is insufficient. This paper did not compare the proposed method with other advanced DNA storage error correction schemes.

7. In the introduction, the reasoning behind the main contributions of the paper is not comprehensively described and highlighted.

8. There is no code available to ensure the reproducibility of our results.

**Questions:**

1. The experiments are conducted solely on MNIST, which is relatively simple and not representative of real DNA storage data. Could the authors clarify whether NeuroDNAAI can generalize to larger or more complex datasets?

2. The PCR-aware module is one of the paper’s most distinctive contributions.
Could the authors clarify how the PCR cycle count and polymerase error profiles were parameterized? Are these parameters derived from real biological data or heuristics?
How sensitive is the model’s reconstruction accuracy to these biological settings?

3. How do the authors envision integrating NeuroDNAAI with existing DNA storage standards or error-correcting code schemes?

---

### Official Review · Reviewer_xfkT · 2025-10-30

**Soundness:** 1
**Presentation:** 1
**Contribution:** 1
**Rating:** 2
**Confidence:** 4

**Summary:**

The paper proposes an end-to-end DNA storage framework based on a Transformer-based neural decoder. Digital information is encoded into DNA sequences, passed through a configurable noise model that simulates synthesis and sequencing errors, and subsequently reconstructed using an encoder–decoder architecture.

**Strengths:**

Thepaper encodes digital information into DNA sequences to achieve higher storage capacity. The encoding design specifically incorporating biochemical constraints such as GC content imbalance and homopolymers, while also accounting for robustness requirements in noisy environments.

**Weaknesses:**

The paper suffers from substandard writing quality and lacks innovation in network architecture and methodology. It fails to provide necessary numerical experiments and discussion, which fails to meet the high standards of ICLR.

**Questions:**

I only list the main issues concerning methodology and experiments.

1. in the abstract, the author mentioned `quantum parallelism concepts', but I could not find the detials throughout the paper.  It appears to be conceptually driven but lacks technical depth.

2. The paper exhibits significant shortcomings in its methodology section, particularly in the lack of detailed description regarding network architecture design and the implementation strategy for ensuring biochemical constraints.

3. The experimental results are presented in a preliminary manner, with only one outcomes provided. The limited evidence fails to demonstrate the method's effectiveness in real-world scenarios, as it lacks critical components such as comparative baseline evaluations and ablation studies to validate the model's architecture.

---

### Meta-Review · Area_Chair_XmWk · 2025-12-10

**Summary:**

This paper addresses an important problem in DNA-based data storage, but all reviewers identify significant deficiencies. The methodology and architecture are insufficiently specified, with unclear or unsupported claims (e.g., “quantum parallelism”) and weak technical rigor. Experimental validation is minimal, limited to MNIST, and lacks baselines, ablations, BER analysis, and evidence of real-world relevance. Novelty over prior DNA storage simulation and neural decoding work is not clearly established. Presentation quality and figure clarity further detract from the submission. In the absence of a rebuttal, these concerns remain unresolved, and the paper does not meet the acceptance standard.

**Reviewer Concerns:**

No rebuttal is provided.

**Reviewer Scores:**

No change is expected since there is no rebuttal.

---

### Decision · Program_Chairs · 2026-01-26

Reject